

# Analysis of twelve genomes of the bacterium *Kerstersia gyiorum* from brown-throated sloths (*Bradypus variegatus*), the first from a non-human host

Dennis Carhuaricra-Huaman[1], Irys H.L. Gonzalez[2], Patricia L. Ramos[2], Aline M. da Silva[1] and Joao C. Setubal[1]

[1] Departamento de Bioquímica, Instituto de Química, Universidade de São Paulo, São Paulo, SP, Brazil
[2] Coordenadoria de Fauna Silvestre, Secretaria do Meio Ambiente, São Paulo, SP, Brazil

Corresponding author
Joao C. Setubal, setubal@iq.usp.br

## ABSTRACT

*Kerstersia gyiorum* is a Gram-negative bacterium found in various animals, including humans, where it has been associated with various infections. Knowledge of the basic biology of *K. gyiorum* is essential to understand the evolutionary strategies of niche adaptation and how this organism contributes to infectious diseases; however, genomic data about *K. gyiorum* is very limited, especially from non-human hosts. In this work, we sequenced 12 *K. gyiorum* genomes isolated from healthy free-living brown-throated sloths (*Bradypus variegatus*) in the Parque Estadual das Fontes do Ipiranga (São Paulo, Brazil), and compared them with genomes from isolates of human origin, in order to gain insights into genomic diversity, phylogeny, and host specialization of this species. Phylogenetic analysis revealed that these *K. gyiorum* strains are structured according to host. Despite the fact that sloth isolates were sampled from a single geographic location, the intra-sloth *K. gyiorum* diversity was divided into three clusters, with differences of more than 1,000 single nucleotide polymorphisms between them, suggesting the circulation of various *K. gyiorum* lineages in sloths. Genes involved in mobilome and defense mechanisms against mobile genetic elements were the main source of gene content variation between isolates from different hosts. Sloth-specific *K. gyiorum* genome features include an IncN2 plasmid, a phage sequence, and a CRISPR-Cas system. The broad diversity of defense elements in *K. gyiorum* (14 systems) may prevent further mobile element flow and explain the low amount of mobile genetic elements in *K. gyiorum* genomes. Gene content variation may be important for the adaptation of *K. gyiorum* to different host niches. This study furthers our understanding of diversity, host adaptation, and evolution of *K. gyiorum*, by presenting and analyzing the first genomes of non-human isolates.

## INTRODUCTION

*Kerstersia gyiorum* is a Gram-negative bacterium classified into the *Burkholderiales* order and *Alcaligenaceae* family (*Coenye et al., 2003*), and has been reported in various human diseases, such as chronic ear, respiratory tract, urinary tract, and limb infections (*Almuzara et al., 2012*; *Pence et al., 2013*; *Ogawa et al., 2016*). *K. gyiorum* has also been isolated from other mammals and insects (*Gupta et al., 2012*; *Wan et al., 2016*; *Dalmutt et al., 2020*; *Shen et al., 2022*), suggesting that it belongs to the commensal microbiota of animals and may act as an opportunistic pathogen. Despite being associated with various diseases, the pathogenic mechanism of *K. gyiorum* is unclear. The genomic analysis of human isolates allowed the identification of putative virulence factors and antimicrobial resistance genes in *K. gyiorum* (*Li et al., 2019*). Strains from non-human hosts may have a different genetic makeup compared to strains of human origin, potentially exhibiting novel metabolic capacities, increased virulence, or zoonotic potential.

The brown-throated sloth (*Bradypus variegatus*) is an arboreal mammal from the Xenarthra superorder distributed throughout Central and South America, inhabiting the Amazon, Caatinga and Atlantic Forest biomes (*Hayssen, 2010*). Sloths have a peculiar lifestyle, spending much of their lives hanging from trees and having the lowest metabolism rate of all mammals (*Kaup, Trull & Hom, 2021*). As with other mammals, the sloth gut microbiome is thought to play an important role in sloth behavioral ecology (*Dill-McFarland et al., 2016*). Additionally, sloths are considered important reservoirs of various zoonotic pathogens including *Leishmania, Trypanosoma cruzi, Anaplasma* spp., *Ehrlichia* spp., and arboviruses such as dengue, mayaro, oropouche, among others (*Catenacci et al., 2018*; *Muñoz García et al., 2019*; *Calchi et al., 2020*; *Sant'Ana et al., 2020*). However, the diversity of the sloth microbiome is largely unexplored.

Another motivation for this work is that anthropogenic activities produce rapid changes in the habitat available to wildlife, intensifying the interaction between wild animals, humans and domestic animals, increasing the risk of pathogen transmission (*Kruse, Kirkemo & Handeland, 2004*; *Lindahl & Grace, 2015*).

In this work, we sequenced and characterized *K. gyiorum* genomes isolated from free-living brown-throated sloth from the Parque Estadual das Fontes do Ipiranga (PEFI) in the city of São Paulo, Brazil and performed the first comparative genomic analysis for this bacterial species.

## METHODS

Portions of this text were previously published as part of the first author master's thesis (https://doi.org/10.11606/D.95.2023.tde-07022024-204134).

### Ethic statements

This study did not include any animal experiments, and only non-invasive samples were collected during sampling procedures. All research was performed in accordance with relevant national (see below) and international guidelines and regulations (*Mellor, Hunt & Gusset, 2015*). Guidelines for handling sloths (*Dünner-Oliger & Pastor-Nicolai, 2017*) were

strictly followed. Animal capture and sample collection (rectal swabs) were conducted between 2015 and 2016 with permission of Instituto Chico Mendes da Conservação da Biodiversidade (ICMBio) and Sistema de Autorização e Informação em Biodiversidade (SISBIO) license number 49627, and approved by the scientific board of Fundação Parque Zoológico de São Paulo. In addition, this research was compliant with Brazilian regulations (Lei Arouca, number 11794, October 8, 2008).

## Sample collection and bacterial isolation

Rectal swabs were obtained from wild brown-throated sloths (*Bradypus variegatus*) for culturable bacterial diversity analysis. Among the cultured bacteria, 12 isolates were identified by 16S rRNA gene sequencing as *K. gyiorum* in samples of 10 sloths.

## Disk diffusion susceptibility test

The Kirby-Bauer disk diffusion test was performed to evaluate the susceptibility of *K. gyiorum* isolates to antimicrobials (*Bauer et al., 1966*). Fifteen different antibiotics were tested: amikacin (AMI), amoxicillin plus clavulanic acid (AMC), ampicillin (AMP), cephalexin (CFE), ceftazidime (CAZ), chloramphenicol (CLO), ceftriaxone (CLO), coxycycline (DOX), enrofloxacin (ENO), gentamicin (GEN), ciprofloxacin (CIP), imipenem (IPM), neomycin (NEO), norfloxacin (NOR), and sulfamethoxazole plus trimethoprim (SUT). The inhibition zones of each isolate were measured and compared to the standard defined by Clinical and Laboratory Standards Institute (*CLSI, 2018*).

## Sequencing of 16S ribosomal RNA gene (16S rRNA)

Bacterial growth was promoted by inoculating the samples on Blood Agar with 5% sheep's blood and Mac Conkey Agar at 36 °C for 48 h. After bacterial growth, the morphology of the colonies was assessed and the different types of colonies were isolated using Tryptone Soy Agar and cryopreserved at −80 °C using Tryptone Soy Broth and 20% Glycerol for further 16S rRNA analysis. DNA was extracted from fresh bacterial colonies using Wizard Genomic DNA Purification Kit (Promega, Madison, WI, USA) and used as template for PCR amplification of the V1–V9 region of 16S rRNA gene with primers 27F (5′AGAGTTTGATCMTGGCTCAG 3′) and 1401R (5′CGGTGTGTACAAGACCC 3′) (*Nübel et al., 1996*). PCR reactions (50 µL) with 30–50 ng of genomic DNA, 2U Taq DNA polymerase (Invitrogen, Waltham, MA, USA) and 0.3 µM of each primer were performed on Veriti 96-Well Fast Thermal Cycler (Applied Biosystems, Waltham, MA, USA) under the following conditions: 3 min at 95 °C, followed by 40 cycles of 2 min at 95 °C, 1 min at 50 °C and 3 min at 72 °C, and a final extension step at 72 °C for 10 min. Aliquots of each reaction were analyzed on agarose gel electrophoresis to confirm the amplicon expected size of ∼1,500 bp. The PCR products were purified using GFX PCR DNA and Gel Band Purification Kit (Merck KgaA, Darmstadt, Germany) and subjected to Sanger sequencing using BigDye Terminator v3.1 Cycle Sequencing Kit (Thermo Fisher Scientific, Waltham, MA, USA) and the above mentioned primers 27F and 1401R or primer 782R (5′ACCAGGGTATCTAATCCTGT 3′) (*Chun, 1995*) on an ABI PRISM 3130XL genetic analyzer (Thermo Fisher Scientific, Waltham, MA, USA).

**Table 1  Summary statistics and accession numbers for *K. gyiorum* genomes.**

| Strain | Country | Host | % CG | no. of contigs | Sequencing coverage | No. of bases | N50 | Completeness | Accession number |
|--------|---------|------|------|----------------|---------------------|--------------|-----|--------------|------------------|
| 1483E | Brazil | Sloth | 62.67 | 24 | 402.2x | 3,790,355 | 304,202 | 99.76 | JALJYH000000000 |
| 155D | Brazil | Sloth | 62.67 | 24 | 398.3x | 3,794,361 | 286,375 | 99.76 | JALJXS000000000 |
| 186F | Brazil | Sloth | 62.69 | 19 | 398.7x | 3,781,143 | 421,928 | 99.53 | JALJYN000000000 |
| 186J | Brazil | Sloth | 62.69 | 15 | 398.2x | 3,781,438 | 426,601 | 99.53 | JAOQMY000000000 |
| 262I | Brazil | Sloth | 62.67 | 23 | 399.5x | 3,807,173 | 299,337 | 99.53 | JALJYO000000000 |
| 2780G | Brazil | Sloth | 62.67 | 22 | 399.2x | 3,807,569 | 386,286 | 99.53 | NZ_JALJXY000000000 |
| 3324E | Brazil | Sloth | 62.69 | 19 | 100.9x | 3,781,341 | 421,905 | 99.53 | JALJYL000000000 |
| 3415D | Brazil | Sloth | 62.66 | 23 | 400.0x | 3,809,345 | 386,286 | 99.53 | JALJYP000000000 |
| 3415G | Brazil | Sloth | 62.67 | 21 | 398.5x | 3,809,165 | 809,377 | 99.53 | NZ_JAOQOW000000000 |
| 381J | Brazil | Sloth | 62.67 | 25 | 402.8x | 3,790,540 | 304,202 | 99.76 | NZ_JALJXX000000000 |
| 4201G | Brazil | Sloth | 62.69 | 20 | 399.4x | 3,781,254 | 316,545 | 99.53 | JALJXQ000000000 |
| 652G | Brazil | Sloth | 62.68 | 22 | 401.1x | 3,770,845 | 466,140 | 99.53 | JALJYI000000000 |
| CG1 | USA | Human | 62.43 | 32 | – | 3,943,087 | 215,440 | 99.53 | NZ_LBNE00000000 |
| KG0001 | Ghana | Human | 62.45 | 17 | – | 3,983,113 | 485,120 | 99.53 | NZ_JANKLF000000000 |
| DSM_16618 | USA | Human | 62.36 | 9 | – | 3,982,678 | 801,770 | 99.53 | NZ_SGWZ00000000 |
| SWMUKG01 | China | Human | 62.43 | 1 | – | 3,945,801 | 3,945,801 | 99.29 | NZ_CP033936.1 |

The obtained sequences were used to assemble the almost full length 16S RNA gene sequences, which were then searched against the Genbank/NCBI database with BLASTn, allowing taxonomic identification of the isolates as *K. gyiorum*.

## Whole-genome sequencing and assemblies

Purified DNA of the 12 *K. gyiorum* isolates was sent to the Joint Genome Institute (JGI) for sequencing using Illumina NovaSeq S4 platform producing 2 × 150 pair-end reads. *De novo* assembly was performed using SPAdes 3.14.1 (*Bankevich et al., 2012*) and then annotated with the Prokaryotic Genome Annotation Pipeline (PGAP) (*Tatusova et al., 2016*). Additionally, we retrieved four genomic sequences of *K. gyiorum* from Genbank public database as of 08 November of 2022. Assembly completeness was assessed with CheckM software (*Parks et al., 2015*). The basic metrics of all sequences used for this study are shown in Table 1.

## Average Nucleotide Identity (ANI) assessment and phylogenomic analysis

The Average Nucleotide identity (ANI) was calculated for all pairs of genome sequences with the aim of determining whether all isolates belong to the same species. We used the python pipeline PyANI v0.2.7 (https://github.com/widdowquinn/pyani) to generate ANIm values using MUMmer for alignment. A heatmap was generated from the pairwise ANI values matrix using the *pheatmap* package in R v3.5.

For phylogenomic analysis, we used Snippy v 4.6.0 (https://github.com/tseemann/snippy) to generate a core genome alignment. The complete genome sequence of *K. gyiorum* str. SWMUKG01 was used as a reference (accession number: CP033936). We used Gubbins v2.4.1 (*Croucher et al., 2015*) to remove recombinant regions prior to the phylogenetic

reconstruction. From the core genome alignment, we reconstructed a maximum-likelihood tree using IQ-TREE v1.6.12 with 1,000 bootstrap replicates and based on a GTR+F+I+R8 nucleotide substitution model as predicted by ModelFinder (*Nguyen et al., 2015*). The ggtree v3.0.4 package was used for visualization and annotation of the phylogenetic tree (*Yu et al., 2017*).

The population structure of the sloth isolates was further defined using RhierBAPS (*Tonkin-Hill et al., 2018*) based on a core genome alignment.

## Pangenome reconstruction, functional annotation, and gene content variation analysis

The annotated genome assemblies were used as input for Panaroo (*Tonkin-Hill et al., 2020*). Orthologous genes were grouped using protein sequence identity of at least 95% and query coverage of at least 70%, which determined the core and accessory genes of 16 *K. gyiorum* genomes. The pangenome was represented on a heatmap using the presence/absence matrix of orthologous genes (OGs). Pan- and core genome curves were generated from the presence/absence matrix of orthologous groups retrieved from Panaroo using the R package vegan.

We used the Clusters of Orthologous Group (COG) database (*Galperin et al., 2021*) for functional annotation, performing a conserved-domain search with RPS-BLAST against COG domains. The command line for RPS-BLAST was rps-blast–query protein.faa -db COG -out protein_out.out -evalue 0.001 -outfmt 6. COG categories were assigned to each ortholog gene of the core and accessory genome. If a gene was assigned to more than one of the 26 COG categories, it was defined as 'ambiguous' category.

We used the presence/absence matrix of accessory genes recovered from Panaroo to perform a principal component analysis (PCA) analysis to assess clustering between isolates from different hosts using ggfortify package in R. BLAST Ring Image Generator (BRIG) v 0.95 (*Alikhan et al., 2011*) was used to align and compare genomes from different hosts. Variable regions identified by BRIG comparison were extracted and manually checked for accurate annotation using EasyFig version 2.2.273 (*Sullivan, Petty & Beatson, 2011*).

## Detection of mobile genetic elements (MGEs), defense mechanism systems, virulence factors and ARGs

To search for phages in the *K. gyiorum* genomes we selected two command-line tools: PhiSpy (*Akhter, Aziz & Edwards, 2012*) and VirSorter2 (*Guo et al., 2021*). Both tools performed well when evaluated for their ability to predict prophages in bacterial genomes (*Roach et al., 2022*). We adopted as predictions only those sequences that were predicted to be phage sequences by both tools.

For plasmid prediction, we use PlasmidFinder (*Carattoli & Hasman, 2020*), whereas Integronfinder2 (*Néron et al., 2022*) and MGEfinder (*Durrant et al., 2020*) were used to identify integrons and other MGEs. Defense systems were detected in the genomes of *K. gyiorum* with DefenseFinder (*Tesson et al., 2022*) using default settings.

ABRICATE (https://github.com/tseemann/abricate) was used to screen the assembly sequences against the CARD and RESFINDER databases for antimicrobial resistance genes (ARGs) detection and against the Virulence Factor Database (VFDB) for virulence factor

identification. We set the percent identity and subject coverage thresholds at 70% and 70%, respectively. The ggtree package was used for plotting a heatmap of presence-absence genes.

## RESULTS

### General features of *K. gyiorum* genomes

We isolated and sequenced 12 strains of *K. gyiorum* obtained from rectal swabs of free-living brown-throated sloths, which were sampled during clinical examination at Fundação Parque Zoológico de São Paulo, between 2014 and 2016. These 12 draft genomes plus four publicly available sequences from the GenBank were analyzed (Table 1). Overall, the genome size of these 16 strains ranged from 3.77 to 3.99 Mbp, with an average length of 3.84 Mbp. The genomic sizes of strains from human origin (3.95–3.99 Mbp) were longer than those strains isolated from sloths (3.77 to 3.81 Mbp) (Fig. S1A). GC content ranged from 62.35 to 62.69%; a small difference is observed between isolates from different hosts (Fig. S1B).

### Phylogenetic analysis reveals a structured set of *K. gyiorum* strains

The ANI values for all pairs of genomes ranged from 98.99 to 99.99%, which means that all 16 genomes belong to *K. gyiorum* species and are closely related, despite differences in genomic size and GC content (Table 1). A maximum likelihood phylogeny was constructed using the core genome alignment of 16 genomes of *K. gyiorum* as well as other members of the *Alcaligenaceae* family: *Alcaligenes faecalis* strain DMS-30030, *Alcaligenes aquatilis* strain QD168 and *Bordetella bronchiseptica* strain NCTC10543, which served as outgroups (Fig. 1A). The phylogenetic analysis showed a host-associated structure in these *K. gyiorum* genomes, with all isolates from sloths grouping in a single clade, separate from human isolates (Fig. 1B).

The 12 *K. gyiorum* isolates sequenced in this study were isolated from 10 brown-throated sloth individuals (individuals' IDs: A, B, C, E, F, G, H, I, J and L). SNP distance and population structure between isolates revealed an important genetic diversity of *K. gyiorum* in sloths at PEFI. Three clusters (B1, B2 and B3) with at least a thousand SNPs of difference between the clusters were observed (Fig. 1C). Two individuals (B and J) provided two isolates each, and their SNP distance was between zero or two SNPs, which suggests that the intra-host diversity of *K. gyiorum* may be the result of a single clone, unlike other commensal bacteria, such as *Escherichia coli*, which maintains multiple lineages within an individual's gastrointestinal microbiota (*Mäklin et al., 2022*).

### The pangenome of *K. gyiorum*

We identified 4,199 different orthologous genes in 16 genomes of *K. gyiorum*, representing its pangenome (Fig. 2A), where 2,974 (70%) genes were shared by all strains (core genome), and 1,225 (30%) genes were present in the accessory fraction (cloud and shell genomes). The pangenome rarefaction curve shows an increasing trend (Fig. 2B). Heaps law allows calculation of whether or not the pangenome is open or closed based on the value of $\alpha$ in the equation $n = \kappa N^{-\alpha}$, where $n$ is the number of genes for a given number of genomes

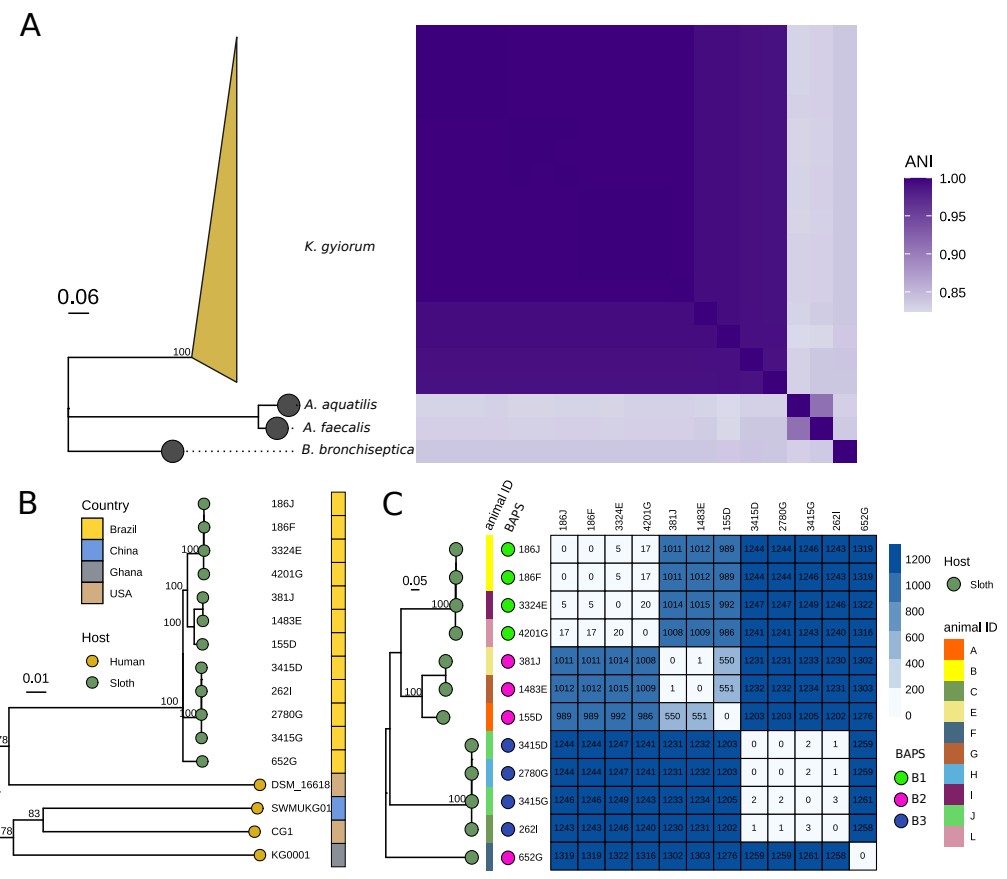

**Figure 1** **Genetic relatedness and phylogenomics of *K. gyiorum* strains.** (A) Phylogenetic tree based on SNP alignment of the core genome of *K. gyiorum* isolates. *Alcaligenes faecalis* DMS-30030 and *Alcaligenes aquatilis* QD168 were used as outgroups, coupled to a heatmap of average nucleotide identity (ANI) values. *K. gyiorum* strains share more than 95% ANIm, confirming they are the same species. (B) Phylogenetic tree of 16 *K. gyiorum* isolates, with host and country information. (C) Population structure of 12 *K. gyiorum* isolates from sloths identified three BAPS clusters (B1-B3), and the SNP distance between isolates is shown in a heatmap. The animal ID (A–L) of each isolate is shown in a colored strip.

($N$) and parameters $\kappa$ and $\alpha$ are calculated in the process of curve fitting (*Tettelin et al., 2008*). In our analysis, $\alpha = 0.91$, which is evidence that the *K. gyiorum* pangenome is open. However, the gene frequency distribution across strains shows a closed pangenome pattern (Fig. 2C), with most genes present in all strains (*Domingo-Sananes & McInerney, 2021*); this can be explained by the low strain diversity (lowest ANI is 98.99%) that we used in this analysis.

## Functional assignation of *K. gyiorum* pangenome fractions

The COG classifications of each gene of the core and accessory genomes are depicted in Fig. 3 (detailed descriptions are presented in Tables S1 and S2). First, 3,124 (74.4%) of all 4,199 genes from the pangenome were assigned to one COG category, while 451 (10.7%) had an ambiguous COG category annotation (more than one COG category), and 622 (14.8%)

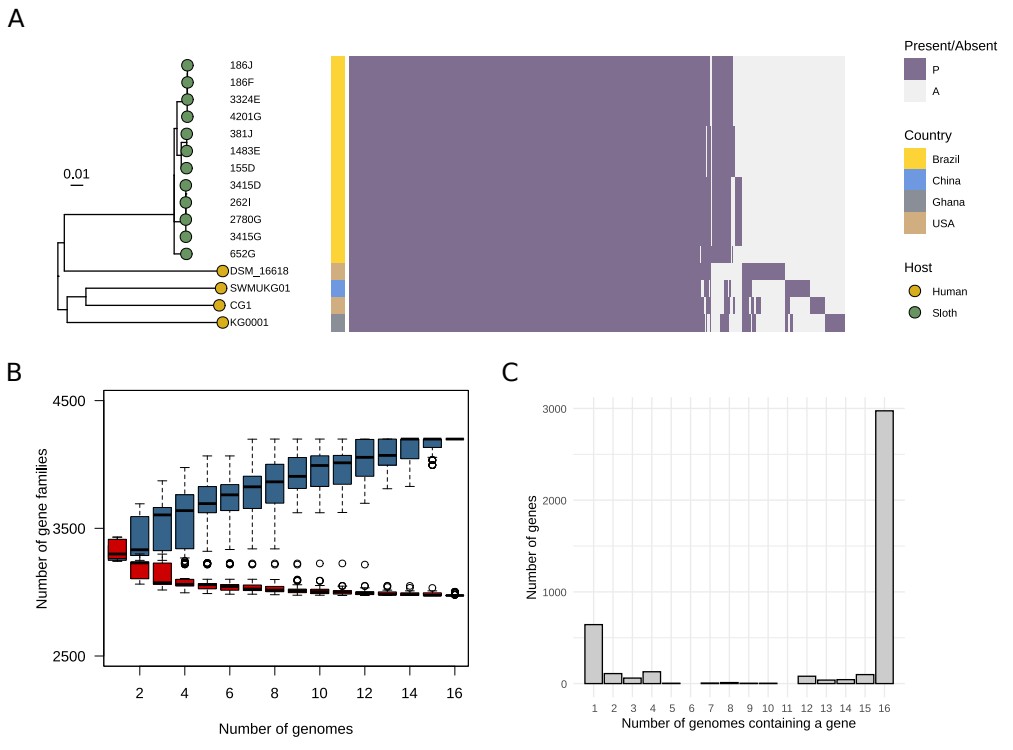

**Figure 2** **Pangenome of *Kerstersia gyiorum*.** (A) Phylogenetic tree of 16 *K. gyiorum* genomes alongside a presence/absence matrix of all 4,199 genes from the pangenome. (B) Curves for pan- (blue) and core- (red) genomes of *K. gyiorum*, as calculated using *vegan* package in R. The power-law regression model suggests that the pangenome is open (=0.9). (C) Gene frequency distribution across strains.

lacked a COG annotation (unclassified). The COG categories more frequently present in the pangenome were K (Transcription), E (Amino acid transport and metabolism), P (Inorganic ion transport and metabolism), J (Translation, ribosomal structure and biogenesis), and M (Cell wall/membrane/envelope biogenesis). These five categories accounted for 7.66, 7.31, 5.24, 5.07 and 4.81% of the *K. gyiorum* pangenome, respectively. The E, K, J, P and I (Lipid transport and metabolism) categories together represent 35.9% of all genes in the core genome (Fig. 3A), whereas categories K, X (Mobilome: prophages, transposons), V (Defense mechanisms) and L (Replication, recombination and repair) are the most frequent in the accessory genome (25.7%). The relative number of genes present in the accessory genome was several-fold higher than in the core genome for the following informative COG assignments: mobilome: prophages, transposons (X, 20.8-fold), Intracellular trafficking, secretion, and vesicular transport (U, 2.44-fold), and Defense mechanisms (V, 2.04-fold).

## Specific gene content variation in sloth and human isolates

Principal component analysis (PCA) of the accessory genome shows that gene content variation discriminates *K. gyiorum* genomes according to host (sloths and humans) (Fig. 4A). We identified 76 genes that were present only in sloth isolates and 54 genes that
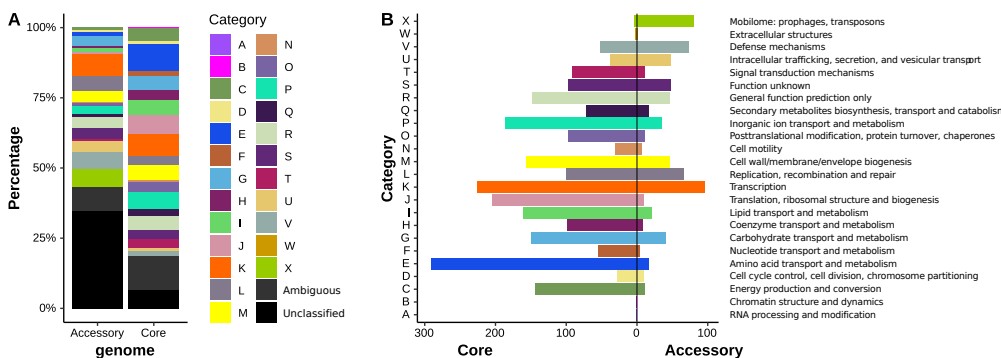

**Figure 3 Functional annotations of *K. gyiorum* core and accessory genomes.** (A) COG categories of genes within the core and accessory genomes of *K. gyiorum*. Each category is graphed as a percentage of the total number of genes in the core or accessory genomes. (B) Back-to-back barplot comparing the functional annotation of core and accessory genes, showing that Mobilome: prophages, transposons (X), Intracellular trafficking, secretion, and vesicular transport (U), and Defense mechanisms (V) categories are more frequent in the accessory than in the core genome.

were present only in human isolates. The functional annotation of these genes revealed different functional assignments (Fig. 4B). While categories I, E, G (Carbohydrate transport and metabolism) and Q (Secondary metabolites biosynthesis, transport and catabolism) are more frequent in genes found only in human isolates, categories X, U (Intracellular trafficking, secretion, and vesicular transport), and V are more frequent in sloth isolates (Tables S3 and S4).

The genes that were found only in the 12 *K. gyiorum* genomes from sloths were arranged in three genomic regions (RS01–RS03) and in an Inc2 plasmid (Figs. 4C and 5B). RS01 contains poorly characterized genes; RS02 contains genes involved in metabolic function: Energy production and conversion (C) and Carbohydrate transport and metabolism (G), while the RS03 region contains CRISPR-associated genes (*cas*) that belong to the Defense Mechanisms (V) category (Fig. 4D). The IncN2 plasmid sequence was predicted to be 21,134 bp long (see below).

We found two regions present in all human isolates but absent in sloth isolates: RH01 and RH02, with 18 and 14 genes, respectively (Fig. S2). For both regions, most genes were annotated with COG categories C (Energy production and conversion), E (Amino acid transport and metabolism), G (Carbohydrate transport and metabolism), and I (Lipid transport and metabolism).

## Mobile genetic elements (MGEs) and defense systems

IncQ1, IncQ2 and IncN2 replicons were detected with at least 70/70% of identity/coverage sequence thresholds; IncN2 and IncQ2 were present in sloth isolates but absent in human isolates (Fig. 5A). The IncQ1 replicon was detected only in the SWMUKG01 strain isolated from humans, but no plasmids were reported in a previous study (*Li et al., 2019*). The contig containing the IncN2 replicon, assembled in different ways in the 12 sloth isolates, has direct repeats at the ends, which is evidence that it represents the complete plasmid
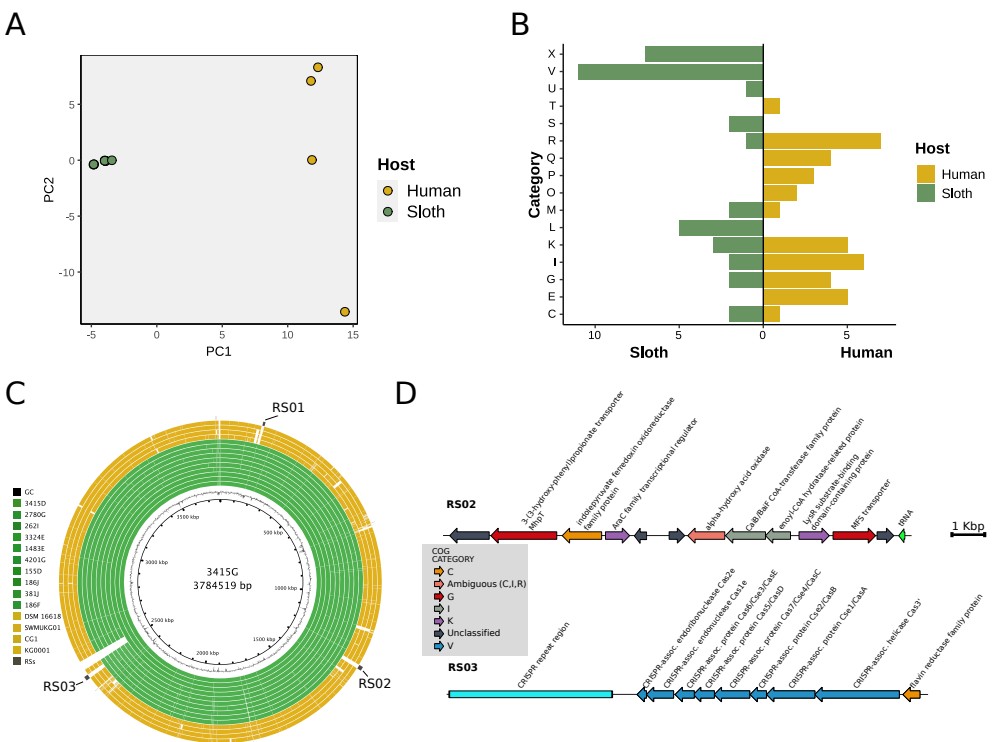

**Figure 4** **Gene content variation between isolates from different hosts.** (A) PCA based on the accessory gene presence/absence matrix. (B) Functional annotation using COG categories to classify specific gene pools reveal different functions in a group of genes of sloth and human isolates. (C) BRIG alignment of all 16 genomes of *K. gyiorum* reveals three genetic regions (RS01-RS03) present in all 12 sloth isolates but absent in human isolates. (D) Gene context of RS02 and RS03 regions of *K. gyiorum* genomes from sloths, colored according to COG category.

sequence; it is 21,134 bp long (without the repeats) (Fig. 5B). A BLASTn search of this sequence against the "nt" database returned a match with 70% subject coverage and 95% identity with the pXap41 plasmid found in the bacterium *Xanthomonas arboricola* pv. *pruni* (*Pothier et al., 2011*). This plasmid carries genes encoding type III effectors and helper genes that are absent in the plasmid of *K. gyiorum*. In contrast, the IncN2 plasmid contained Type IV secretion system genes (*virB6*, *virB5*) and other involved in pili formation, plasmid conjugation (*tra*F, G, H, L and J) and segregation (*par*A and *par*B) (Fig. 5B).

Twenty-one phage sequences were predicted (Table S5). Nine were predicted by VirSorter2 and 12 by PhiSpy. We considered as phages only sequence regions that were predicted by both tools, resulting in five different phage sequences (Table S6). We gave these the following provisional names: *K. gyiorum* phage Kgϕ1 through Kgϕ5 (Figs. 5A, 5C). Kgϕ1 (27.7 Kbp) was the only phage detected in sloth isolates (in four isolates). Each of the four human isolates has a different phage sequence (Kgϕ2–Kgϕ5). All phages were predicted to be dsDNA phages. We did not detect any integrases, transposases or insertion sequences (IS) in the *K. gyiorum* genomes analyzed here.

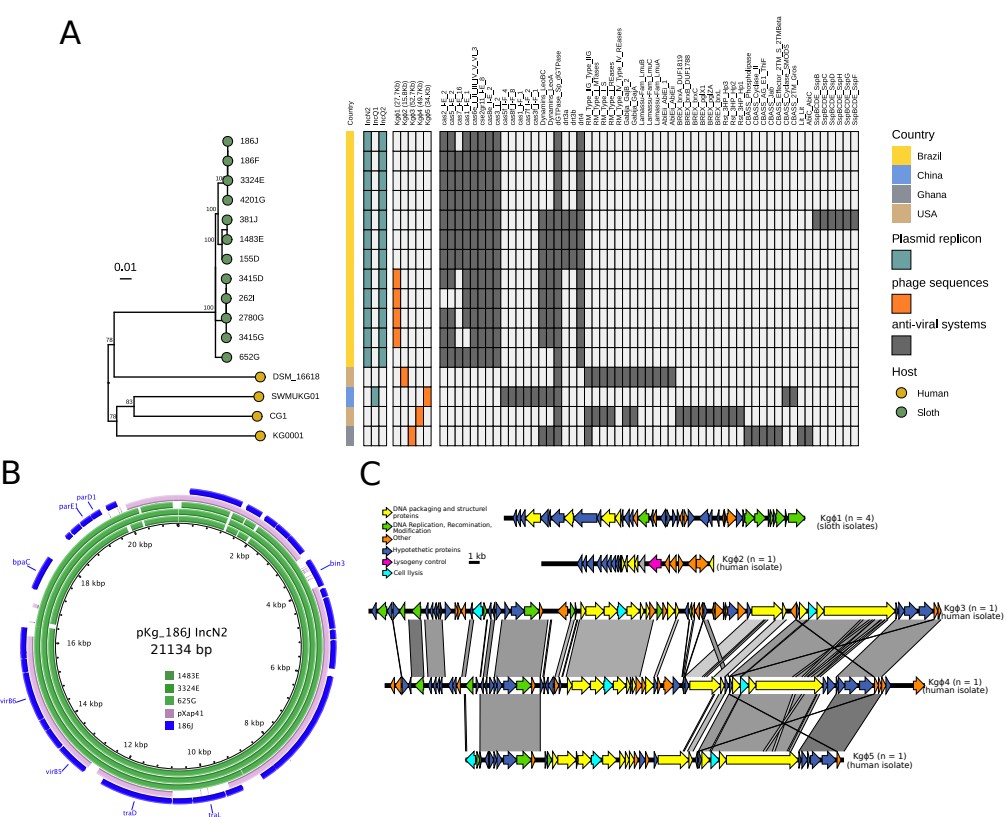

**Figure 5 Mobile genetic elements and Defense Systems in *K. gyiorum*.** (A) Presence/absence matrix of plasmid replicons, phage sequences, and defense systems predicted for the 16 *K. gyiorum* genomes. (B) Comparison of the IncN2 plasmid 21,134 bp sequence of *K. gyiorum* from sloths with the pXap41 plasmid sequence from plant pathogen *Xanthomonas arboricola* pv. *pruni* shows partial alignment (70% of coverage) between these sequences. (C) Map of five predicted phage sequences in *K. gyiorum* genomes.

Microbes can defend themselves against phages and other MGEs using a variety of systems. We detected 14 different defense systems, including cas and restriction-modification (RM) systems (Fig. 5A and Table S6). In all sloth isolates two CRISPR-cas type I operons were predicted, one of them including CRISPR spacers (Fig. 4C). Additionally, other systems were present exclusively in sloth isolates: the defense retro-transcriptase (DRT) system (drt3ab and drt4) and dynamins (LeoA and LeoBC), both involved in anti-viral defense functions. More diverse defense mechanisms were found in human isolates, including RM type I, II and IV; BREX, CBASS, Rst, and Gabija.

## Virulence factors and antimicrobial resistance genes

We found 51 putative virulence factors in *K. gyiorum* genomes. These genes are associated with adhesion, biofilm formation, capsule, flagella, LPS, iron uptake and secretory systems (Fig. 6). Most of these sequences are conserved in isolates from both hosts. The flagellar regulon of *K. gyiorum* was described by *Li et al. (2019)* in the SWMUKG01 genome. Here we have identified operons *cheARWY*, *flgCDEFGHIK*, *flhABC*, *fliFGIMPORS*, and *motAB*,

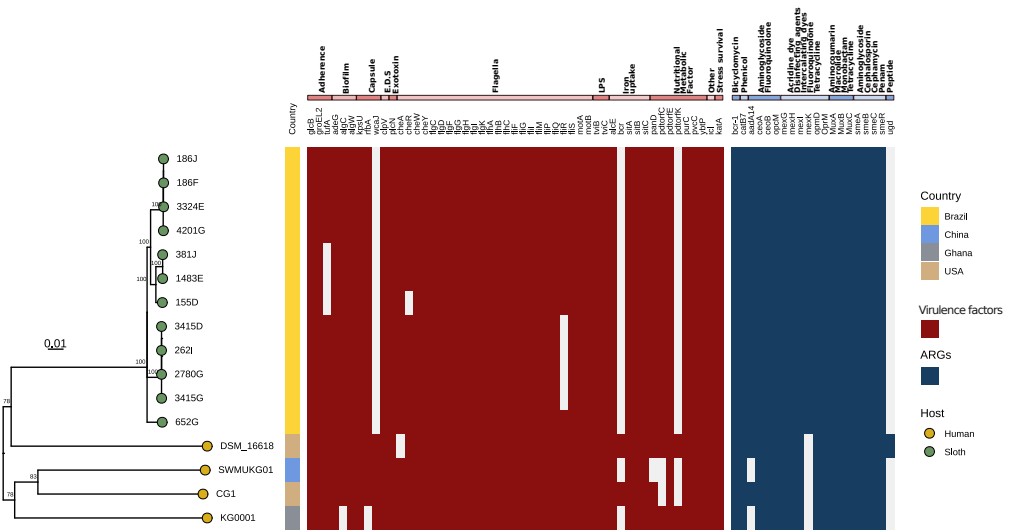

**Figure 6** **Virulome and Resistome of *K. gyiorum*..** Phylogenetic tree of 16 *K. gyiorum* genomes alongside a presence/absence graphic of virulence factors (red) and antimicrobial resistance genes (blue), predicted by ABRICATE using VFDB and CARD databases, respectively.

found in all isolates with the exception of *fliR*, which was missing in four sloth isolates. Cell surface components like capsule and lipopolysaccharides (LPSs) are essential virulence factors in some Gram-negative bacteria. We identified a highly conserved operon *tviBC*, whose gene products are involved in the biosynthesis of LPS and capsule in pathogenic bacteria like *Salmonella enterica* serotype Typhi and *Bordetella bronchiseptica* (*Wear et al., 2022*). The *wcaJ* gene, whose product is related to capsule formation, was only identified in human isolates. On the other hand, five genes, whose products are associated with iron uptake, including the *sitABC* operon, which is involved in iron uptake in *Staphylococcus* (*Massonet et al., 2006*), were found in all *K. gyiorum* genomes analyzed here. The *bcr* gene (*alcS* in *Bordetella spp.*) was identified only in three human isolates and encodes a permease that maintains appropriate levels of cellular alcaligin levels in *Bordetella* species (*Alcaligenaceae* family) (*Brickman & Armstrong, 2005*).

Phenotypic characterization of antimicrobial resistance in sloth isolates using disk diffusion susceptibility testing revealed susceptibility to almost all 15 antibiotics tested; resistance to norfloxacin (quinolone) was observed in two isolates (Table S7). Genomic inspection using the CARD database detected 20 chromosomal genes putatively encoding multidrug efflux pump systems against fluoroquinolones, tetracycline, phenicols, cephalosporins, as well as other antimicrobials (Fig. 6). These included the *ceoAB-opcM* encoded efflux pump with resistance to aminoglycoside and fluoroquinolone; *mexGHI-opmD* encoding resistance to fluoroquinolone and tetracycline, *muxABC-opmB* efflux pump encoding resistance to aminocoumarin, macrolide, monobactam and tetracycline; and the complex *smeABC*, encoding resistance to aminoglycoside and cephalosporin.

## DISCUSSION

*Kerstersia gyiorum* has been isolated from multiple hosts, including mammals and insects, and is associated with infections in humans (*Pence et al., 2013*; *Ogawa et al., 2016*). However, only *K. gyiorum* genomes of clinical origin have been previously described (*Li et al., 2019*). Here we isolated and sequenced 12 *K. gyiorum* strains from healthy free-living brown-throated sloths and performed a comprehensive comparative analysis to other *K. gyiorum* genomes in order to gain insights into the genetic diversity, phylogeny, and the genetic basis of host association of this bacterial species.

Our analyses show substantial differences between the *K. gyiorum* genomes isolated from sloths and humans at various levels. The phylogenomic analysis shows that sloth isolates cluster separately from human isolates, suggesting that *K. gyiorum* diversity is structured according to host. Additionally, we observed differences in genome size and GC content between isolates from both hosts. This pattern may be evidence of host-adapted lineages of *K. gyiorum*, but other explanations are also possible. For example, physical barriers in different host niches followed by genetic drift could create the same distribution (*Sheppard, Guttman & Fitzgerald, 2018*). It is important to remark that the geographic distribution of strains may account for the phylogenetic structure obtained here, since all isolates from sloths were from Brazil.

Sloths have a peculiar lifestyle and physiological adaptations that make their digestive system a unique environment (*Kaup, Trull & Hom, 2021*). Some previous works have shown that the gut microbial population in sloths shows a distinctive diversity compared to other herbivores (*Delsuc et al., 2014*; *Dill-McFarland et al., 2016*). A niche significantly different from that of the human digestive system would favor fixation of specific adaptive signatures. Analysis of other isolates from different locations and hosts would be necessary to assess this hypothesis about *K. gyiorum*.

Host-specific genetic content in *K. gyiorum* isolates can be reflective of the resident host microbiome and their specific gene pool (*Moura De Sousa, Lourenço & Gordo, 2023*). Part of these accessory genes might provide significant benefits by conferring novel metabolic capacities, resistance to stress or virulence contributing to the host adaptation (*McInerney, McNally & O'Connell, 2017*; *Iraola et al., 2017*). The genes identified exclusively in sloth isolates display putative functions involved in host-pathogen interaction. For example, the plasmid IncN2 contains genes of the type IV secretion system (T4SS). This system is responsible for the delivery of effectors in host cells, acquisition of virulence traits, or the killing of neighboring bacteria for niche occupation (*Gonzalez-Rivera, Bhatty & Christie, 2016*). The diversity of T4SS systems in *Bartonella* contribute to the host adaptation (*Wagner & Dehio, 2019*). A type-I CRISPR-Cas cassette was also specific to sloth isolates. The CRISPR-cas systems functions as an antiviral defense and in the regulation of bacterial virulence (*Hatoum-Aslan & Marraffini, 2014*). These systems also regulate the acquisition of MGEs, which could contribute to a better adaptability, as suggested for the *Bacillus cereus* group (*Zheng et al., 2020*). On the other hand, the genes specific to human *K. gyiorum* isolates were more frequently annotated with functions related to carbohydrate and aminoacid metabolism. These genes may contribute to the expansion

of the metabolic capabilities as a strategy for niche adaptation and might be critical for a successful infection (*Rohmer, Hocquet & Miller, 2011*). Thus, the accessory genomes of sloths and human *K. gyiorum* isolates harbor distinct gene sets that are potentially important for host–pathogen interactions, and likely reflect the adaptations to different hosts.

We hypothesize that different *K. gyiorum* populations can be found in sloths of PEFI, as revealed by SNP distance and genetic content. The 12 *K. gyiorum* isolates from 10 sloth individuals were distributed in three clades, which differ between them by more than 1,000 SNPs. In other bacterial species, this genetic distance is indicative of separate lineages. For example, in *Salmonella enterica* ser. Typhimurium, a threshold of approximately 400–600 SNPs is used to define different strains (*Branchu, Bawn & Kingsley, 2018*). On the other hand, genes related to plasmid and phage sequences are the primary source of variation in gene content in sloth isolates (Fig. S3), suggesting that horizontal gene transfer (HGT) has played a role in *K. gyiorum* diversification in sloths.

Although one plasmid and phage sequences were identified in some *K. gyiorum* genomes, our analysis revealed few MGEs in the *K. gyiorum* genomes analyzed, with no presence of transposons, insertion sequences (IS) or integrons detected. A previous work has shown that MGEs in some members of *Alcaligeneaceae* family are abundant (*Ellabaan et al., 2021*). For example, *Achromobacter* spp. and *Bordetella bronchiseptica* contain large quantities of MGEs in their genomes, associated with ARGs (*Ellabaan et al., 2021*). The limited number of MGEs in *K. gyiorum* genomes might be related to the presence of multiple defense mechanisms. We observed that the sloth-isolated genomes had different anti-MGE defense systems as compared to the human-isolated genomes. CRISPR-Cas and RM systems are the most widespread defense systems in prokaryotes (*Tesson et al., 2022*). In sloth isolates the most prevalent system was CRISPR-cas (Type I E). In contrast, RM systems (Types I, II, and IV) were the most common in human isolates.

We identified 51 genes encoding virulence factors, annotated as related to adhesion, biofilm formation, capsule, flagella, LPS, iron uptake, and the secretory system, and almost all of them are present in isolates from both hosts. *wcaJ* (capsule), *bcr* (iron uptake) and *pdtorfk* (metabolism) were only detected in human isolates. These factors can play a role in pathogenesis: *wcaJ* encodes a glucosyltransferase, and experimental deletion of this gene has been associated with diminished virulence and phage susceptibility in *Klebsiella pneumoniae* (*Cai et al., 2019*), while *bcr* is involved in iron uptake regulation in pathogenic *Bordetella* species (*Brickman & Armstrong, 2005*). *K. gyiorum* has been reported as having variable motility (*Coenye et al., 2003*; *Almuzara et al., 2012*); however, the flagellum regulon was conserved in all *K. gyiorum* genomes, whereas *K. similis*, another species of the genus, lacks flagella (*Vandamme et al., 2012*). In *Bordetella bronchiseptica*, motility and/or flagella play an important role during infection, and recently this was also demonstrated for *B. pertussis* (*Hoffman et al., 2019*).

In the genomes analyzed, we identified a group of genes encoding efflux pump systems that potentially may be involved in aminoglycoside, tetracycline, and phenicol resistance (*Poole, 2005*). However, disk diffusion susceptibility testing in sloth isolates revealed susceptibility to almost all antibiotics, suggesting that the predicted genes may not be

involved in resistance. Resistance to norfloxacin (quinolone) was observed in only two isolates. In *K. gyiorum* human isolates, resistance to ciprofloxacin (quinolone) was reported in different countries, including Brazil (*Pence et al., 2013*; *Li et al., 2019*; *Pires et al., 2020*). The main mechanism of fluoroquinolone resistance in Gram-negative bacteria is the accumulation of mutations in DNA gyrase and DNA topoisomerase IV (*Bush et al., 2020*). There is no information on *gyrAB* mutations that may confer resistance against fluoroquinolones in *K. gyiorum*, and we did not find polymorphisms in the alignment of these genes in our dataset. The low levels of resistance to fluoroquinolones reported in *K. gyiorum* may be due to efflux proteins as was observed in *Enterobacteriaceae* (*Poole, 2000*). A standardized CLSI criterion for antimicrobial susceptibility testing and interpretation of results for *K. gyiorum* is needed for a correct understanding and determination of resistance in this bacterium.

We constructed the pangenome for *K. gyiorum*, and the evidence we obtained suggests that it is an open pangenome. The open or closed nature of a pangenome is bound to the lifestyle of the bacterial species (*Tettelin et al., 2008*). An open pangenome indicates that species have a high capacity to exchange genetic material and also indicates a free-living lifestyle with metabolic versatility (*Rouli et al., 2015*; *McInerney, McNally & O'Connell, 2017*). This definition is appropriate to the *K. gyiorum* lifestyle, which is found in multiple animal hosts. However, our comparative analysis was limited by the availability of genomes from only two mammal hosts (human and sloth), and therefore the genomes we have analyzed are not representative of the full diversity of this species. The number and diversity of genomes used in the analysis impact measurements of pangenome openness (*McInerney et al., 2020*). Therefore, it is desirable to increase the number of *K. gyiorum* sequences from diverse hosts and geographic settings to gain a better understanding of its genomic diversity and pangenome.

## CONCLUSIONS

Here we presented the first genomic sequences of *K. gyiorum* isolated from animals and performed a comprehensive comparative genomic analysis for this bacterial species. Our phylogenetic analysis showed that the two groups of *K. gyiorum* genomes separate according to host, suggesting that these lineages may be host-specific. An open pangenome was determined for the species in concordance with *K. gyiorum* versatility. We observed few MGEs, likely because of the high diversity of defense mechanism systems. Differences in gene content in isolates with different functional assignations suggest differences in metabolic capabilities in lineages from different hosts.

The main limitations of this study are the small number of genomes analyzed and the fact that the two groups that were compared are rather different (isolates from healthy sloths in Brazil and isolates from human patients in other parts of the world). These limitations need to be taken into consideration regarding the conclusions drawn from our analyses. Nevertheless, the very scarcity of *K. gyiorum* genomes implies that the publication of twelve new genomes of this species is of value to the research community, despite these limitations.

## ACKNOWLEDGEMENTS

The authors thank all the biologists and veterinarians involved in Bradypus Research Program for their support in capturing the animals and collecting samples. Sequencing, assembly, and annotation of the 12 *Kerstersia* genomes were performed at the Joint Genome Institute (JGI) within the framework of the Genomic Encyclopedia of Type Strains project, phase IV. We thank Nikos Kyrpides, Nicole Shapiro, and Natasha Brown at JGI, who enabled the genome sequencing and facilitated the deposition of the data into the IMG/JGI platform. We would like to thank William B. Whitman for advice on selecting the *K. gyiorum* strains to be sequenced and for suggestions that helped improve a first version of the manuscript.

### Funding

The work (proposal: 10.46936/10.25585/60001079) was conducted by the U.S. Department of Energy Joint Genome Institute, a DOE Office of Science User Facility, supported by the Office of Science of the U.S. Department of Energy operated under Contract No. DE-AC02-05CH11231. This work was also funded by grant 2011/50870-6 from the São Paulo State Research Foundation (FAPESP). Aline M da Silva and Joao C Setubal received Research Fellowship Awards from the National Council for Scientific and Technological Development (CNPq). There were no other external funding sources. The funders had no role in study design, data collection and analysis, decision to publish, or preparation of the manuscript.

### Grant Disclosures

The following grant information was disclosed by the authors:
The U.S. Department of Energy Joint Genome Institute.
A DOE Office of Science User Facility, was supported by the Office of Science of the U.S. Department of Energy: DE-AC02-05CH11231.
The São Paulo State Research Foundation (FAPESP): 2011/50870-6.
The National Council for Scientific and Technological Development (CNPq).

### Competing Interests

João C. Setubal is an Academic Editor for PeerJ.

### Author Contributions

- Dennis Carhuaricra-Huaman performed the experiments, analyzed the data, prepared figures and/or tables, authored or reviewed drafts of the article, and approved the final draft.
- Irys H.L. Gonzalez conceived and designed the experiments, performed the experiments, analyzed the data, authored or reviewed drafts of the article, and approved the final draft.
- Patricia L. Ramos conceived and designed the experiments, performed the experiments, analyzed the data, authored or reviewed drafts of the article, and approved the final draft.
- Aline M. da Silva conceived and designed the experiments, performed the experiments, analyzed the data, authored or reviewed drafts of the article, and approved the final draft.
- Joao C. Setubal conceived and designed the experiments, performed the experiments, analyzed the data, prepared figures and/or tables, authored or reviewed drafts of the article, and approved the final draft.

### Animal Ethics

The following information was supplied relating to ethical approvals (i.e., approving body and any reference numbers):

This study did not include any animal experiments, and only non-invasive samples were collected during sampling procedures. All research was performed in accordance with relevant national (see below) and international guidelines and regulations (*Mellor, Hunt & Gusset, 2015*). Guidelines for handling sloths (*Dünner-Oliger & Pastor-Nicolai, 2017*) were strictly followed. Animal capture and sample collection (rectal swabs) were conducted between 2015 and 2016 with permission of Instituto Chico Mendes da Conservação da Biodiversidade (ICMBio) and Sistema de Autorização e Informação em Biodiversidade (SISBIO) license number 49627, and approved by the scientific board of Fundação Parque Zoológico de São Paulo. In addition, this research was compliant with Brazilian regulations (Lei Arouca, number 11.794, October 8, 2008).

### Data Availability

The sequencing for this work was done at the Joint Genome Institute (USA), which has the raw data (reads). The 12 *Kerstersia gyiorum* genomes are available at GenBank: JALJYH000000000, JALJXS000000000, JALJYN000000000, JAOQMY000000000, JALJYO000000000, NZ_JALJXY000000000, JALJYL000000000, JALJYP000000000, NZ_JAOQOW000000000, NZ_JALJXX000000000, JALJXQ000000000, JALJYI000000000.

### Supplemental Information

Supplemental information for this article can be found online at http://dx.doi.org/10.7717/peerj.17206#supplemental-information.

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
