# Peer review of "Analysis of twelve genomes of the bacterium Kerstersia gyiorum from brown-throated sloths (Bradypus variegatus), the first from a non-human host"

_PeerJ, doi:10.7717/peerj.17206_

## Round 0.1 · original submission · Major Revisions

Dear Dr. Carhuaricra-Huaman and colleagues:

Thanks for submitting your manuscript to PeerJ. I have now received two independent reviews of your work, and as you will see, the reviewers raised some concerns about the manuscript. Despite this, these reviewers are optimistic about your work and the potential impact it will have on research studying Kerstersia gyiorum biology and evolution. Thus, I encourage you to revise your manuscript, accordingly, considering all the concerns raised by both reviewers.

While the concerns of the reviewers are relatively minor, this is a major revision to ensure that the original reviewers have a chance to evaluate your responses to their concerns (if deemed necessary). There are helpful suggestions, which I am sure will greatly improve your manuscript once addressed.

Please ensure that all missing information noted by the reviewers is included in your revision. Also, please elaborate more on your findings in the Discussion.

I look forward to seeing your revision, and thanks again for submitting your work to PeerJ.

Good luck with your revision,

-joe

**Language Note:** PeerJ staff have identified that the English language needs to be improved. When you prepare your next revision, please either (i) have a colleague who is proficient in English and familiar with the subject matter review your manuscript, or (ii) contact a professional editing service to review your manuscript. PeerJ can provide language editing services - you can contact us at [email protected] for pricing (be sure to provide your manuscript number and title). – PeerJ Staff

·

Basic reporting

Data availability:
The sequence data in GenBank is uploaded with very limited metadata, e.g. "host: not applicable" and
"strain collected from animal source". As a minimum, the authors should add information about isolation country, year, host species and host health status. This will make the data far more useful for the research community.

Table 1: I would add average coverage for the isolates sequenced by the authors.

Line 212: Check consistent use of comma to separate thousands

Line 288: Typhi?

Line 297: "almost all"?

Experimental design

General: I believe DSM 16618 is the same strain as CCUG 47000, under codes from two culture collections. Data from both should not be included in the comparison.

Line 82: Presumably this required selective culture of some kind? Briefly describe how colonies were grown and selected for 16S confirmation.

Validity of the findings

Lines, 26-27, 187-188, 314-315, 423-424, 431-433, elsewhere:

The link between genome evolution and host association is interesting, and the authors have provided valuable data by characterising non-human isolates of Kerstersia. However, they are not suited for drawing conclusions as the groups being compared are small and differ not only in their host species but also in their country of origin and host health status. Essentially, isolates from healthy sloths in South America are being compared to isolates from symptomatic cases in humans on other continents. This is briefly discussed in the current text but it should be stated more clearly.

I would also recommend a different title, the presented dataset is insufficient for identifying distinct features of one host or the other.

Additional comments

In general, this is high quality work of possible interest for both understanding zoonotic infections and the evolution of host bias. Wildlife are generally under-sampled, making the work particularly valuable.

·

Basic reporting

Basic reporting

Resume: The manuscript provides a genome comparison of Ketersia gyiorum isolates recovered from sloths. The genomic analysis reveals distinctive features when compared to human isolates. In addition to genomic analysis, the authors conducted phylogenetic analysis, gene content analysis, antimicrobial resistance (ARG) and virulence gene prediction, and mobile genetic elements (MGE) identification.

+++ Clear and unambiguous, professional English used throughout: Yes

+++ Literature references, sufficient field background/context provided: Yes

+++ Professional article structure, figures, tables. Raw data shared:
Revision:

Abstract:
--- Line 25: In the abstract, the authors refer to "evolutionary history." However, the manuscript only includes phylogenetic analysis and SNP calling. A sole phylogenetic analysis does not constitute an analysis of evolutionary history. To comprehensively evaluate the evolutionary pathway, the authors should assess additional parameters such as dN/dS, Diversity Indexes, Dating, Selection Tests, etc. I recommend removing the term "evolutionary history."

Introduction:
--- Line 49: Replace "wild host" with "non-human host."

Methods:
--- Line 131: How was the substitution model predicted? Was ModelFinder used? The authors should clarify this point.
--- Line 144-148: The authors used RPS-BLAST for COG category assignment. What RPS-BLAST parameters were employed for the analysis?
--- Line 155-169: The authors used a specific database and BLAST to predict different groups of genes (ARG, Virulence, etc.). Why was the parameter of 70% identity and 70% coverage chosen? Additionally, for gene prediction, an e-value cutoff could be considered.

Results:
--- Line 236-242: In Figure 4C and the manuscript, the authors identified four regions. The region RS04 is labelled as a plasmid sequence, but the figure indicates that this region is part of the chromosome, causing confusion. If RS04 is indeed a plasmid sequence, the author should eliminate any indication in Figure 4C. Additionally, the text should accurately describe RS04 as a plasmid.

Discussion:
The discussion is excessively long and challenging to read. Some points are merely comments on results without a connection to the reference literature. The discussion requires revision and reformulation by the authors to ensure coherence and alignment with existing literature.

+++ Self-contained with relevant results to hypotheses: Yes

Experimental design

+++ Original primary research within Aims and Scope of the journal: Yes

+++ Research question well defined, relevant & meaningful. It is stated how research fills an identified knowledge gap: Yes

+++ Rigorous investigation performed to a high technical & ethical standard: Yes

+++ Methods described with sufficient detail & information to replicate: In basic reporting, I indicate the detail and information necessary to replicate the results.

Validity of the findings

No comment

---

## Round 0.2 · accepted · Accept

Dear Dr. Carhuaricra-Huaman and colleagues:

Thanks for revising your manuscript based on the concerns raised by the reviewers. I now believe that your manuscript is suitable for publication. Congratulations! I look forward to seeing this work in print, and I anticipate it being an important resource for groups studying Kerstersia gyiorum biology and evolution. Thanks again for choosing PeerJ to publish such important work.

Best,

-joe

·

Basic reporting

-

Experimental design

-

Validity of the findings

-

Additional comments

The authors have addressed the points raised in my initial review, and I am happy recommend acceptance of the revised version of the manuscript.